# What are the barriers to, and enablers of, working with people with lived experience of mental illness amongst community and voluntary sector organisations? A qualitative study

**Louise Baxter**, **Daisy Fancourt** *

Department of Behavioural Science and Health, University College London, London, England, United Kingdom

* d.fancourt@ucl.ac.uk

## Abstract

There is increasing emphasis on psychological and social approaches to managing and treating mental illness, including a growing evidence base on the effectiveness of community-based social interventions including arts and heritage activities, library programmes, volunteering schemes, nature-based activities and community groups. However, there is a gap in understanding of what the barriers to, and enablers of, working with individuals with mental illness might be for the community and voluntary sector. A qualitative approach was used involving focus groups with non-profit organisations delivering social activities within communities across the United Kingdom. Behaviour Change Theory, the COM-B model and the Theoretical Domains Framework, were employed as the theoretical framework, to develop interventions to address the barriers raised. Representatives of the organisations reported being motivated by the mental health needs of others, and by seeing the benefits of participation. Further motivations included expanding inclusion, and economic motivation to ensure sustainability. Strengths identified included offering innovative, responsive services that were distinct from conventional mental health services. Running these services demanded new and potentially challenging skills, such as understanding statutory responsibilities, and being able to train and support staff. Further challenges included maintaining boundaries between their roles as community organisations and clients' mental health needs and avoiding burnout. Ability to deliver this work was enhanced by support of peer organisations and opportunities to share practice. However, funding was often short term, and complex to obtain, which could destabilise organisations' sustainability. Lack of transparency around the process, differences in language between the community and health sectors, and confusion around commissioning pathways undermined the potential opportunity offered by social prescribing policy. Interventions to address these barriers were identified, including long term funding to support core costs, training on engaging with the commissioning process, around mental health support and safeguarding, and developing mentoring schemes and local co-operatives of organisations for developing partnerships with the health sector.

**Data Availability Statement:** Data cannot be shared publicly because ethical approval only covered analysis for this study as there is a substantial risk of identification of participants.

Data are available from the UCL Ethics Committee (contact via Louise Baxter l.baxter@ucl.ac.uk) for researchers who meet the criteria for access to confidential data.

**Funding:** This work was supported by UK Research and Innovation (UKRI https://www.ukri.org/) as part of the MARCH Mental Health Research Network [ES/S002588/1] (DF) Further support was provided by the Leverhulme Trust (https://www.leverhulme.ac.uk/) [PLP-2018-007] (DF) The funders had no role in study design, data collection and analysis, decision to publish, or preparation of the manuscript.

**Competing interests:** The authors have declared that no competing interests exist.

# Introduction

There is increasing emphasis on integrating biomedical and psychological approaches to managing and treating mental illness with social approaches [1]. Indeed, there is a growing evidence base showing the effectiveness of community-based interventions from arts activities to volunteering to nature-based interventions to social groups for reducing symptoms of mental illness and improving positive psychological factors, such as life satisfaction and mental wellbeing [2–5]. For example, attending community choirs, or postnatal singing groups, have been found to improve symptoms of mental distress, and post-natal depression symptom severity, respectively [6,7]. Engaging with these community activities is also associated with improvements in subjective well-being in older adults [8], whilst regular community cultural engagement is associated with a reduction in depression in older adults [9]. Additionally, increased sense of community, social connection, sense of purpose, and resilience have been reported from studies exploring the impact of community projects as diverse as community gardening, rural arts groups, and, for older people, participating in community theatre [10–14].

In particular, community-based interventions have become popular in treating individuals with mild or moderate mental illness where (in contrast to more severe cases) medication is not always effective [15–19]. For more severe mental illness, when community-based interventions are offered alongside traditional medical treatments, improvements have been found both in mental health symptoms and also in coping skills, which can positively influence the effects of medication [20,21], and support in the maintenance of recovery [20]. Further, integrating community-based interventions as part of a portfolio of options for individuals experiencing mental illness is a way of providing more resource to meet the rising demand for mental health services, which has been precipitated by a variety of influences, including workforce shortages, social factors such as unemployment, and welfare restructuring under austerity policies.

However, what remains unclear is whether the community and voluntary sector (CVS) is prepared for the increasing demand for community activities to support individuals with mental illness. Whilst there has been significant grass-roots enthusiasm for delivering programmes focused on improving health or in partnership with healthcare providers in the past two decades, at present, there is a gap in understanding the possible barriers and enablers that might be faced by community and voluntary sector organisations (CVSOs) in engaging with individuals with mental illness [22]. This is important, as research suggests that individuals with mental illness are less likely to engage voluntarily in community activities due to issues such as perceived stigma [23]. As a result, developing provision of activities for individuals with mental illness can include CVSOs developing partnerships with health or social care professionals through schemes such as social prescribing [24–26], or developing bespoke programmes or bespoke communication of existing programmes, all of which require additional work and expertise from CVSOs.

There is some preliminary data to suggest that there could be important barriers for CVSOs specifically relating to their engagement with social prescribing. An evaluation of a discrete social prescribing scheme mentioned the importance of a shared vision and language across primary care and the CVSOs they are working with, relationship and partnership building, and the importance of a CVS infrastructure to support this [27]. Similarly, case studies of partnerships between GPs and the CVS have identified clashes of culture and 'negative perceptions' internally, in addition to external challenges including lack of funding, unclear referral pathways, and high staff turnover [28,29]. Factors that were mentioned as important for strengthening the relationships between the providers were 'mutual respect' and equality within the relationship, and 'early stakeholder engagement' [28,29]. However, there are still major gaps in understanding these barriers, including around identifying the barriers faced by

organisations in undertaking work with individuals with mental illness outside of formal social prescribing schemes.

Therefore, to extend this preliminary work, this study used the lens of behaviour change theory and applied the COM-B model, which proposes that behaviour is influenced by capabilities (C) to carry the behaviour encompassing both physical skills, and psychological knowledge; opportunities (O) to carry out the behaviour afforded by the physical and social environment; and motivation (M) to undertake the behaviour (B), whether the motivation is reflective (such as planning, goals and intentions) or automatic (including emotional responses) [30]. In seeking to investigate these factors, this study involved in-depth focus groups with CVSOs to identify the factors that facilitate (enable) or act as barriers to the involvement of CVSOs in working with individuals with mental illness.

## Methods

### Design

A qualitative approach was used in order to understand what those working in CVSOs consider to be the barriers to working with people with lived experience of mental illness. We focused both on the work of CVSOs with social prescribing schemes that involve referrals for individuals with mental illness from health, social care or educational professionals to community activities [31], but also on wider activities that CVSOs might be providing in the community for individuals with mental illness. Focus groups were selected in order to provide the opportunity to bring together CVSOs to share with each other some of the work they were already engaged in, good practice that had been developed, and concerns that they had about the work. It was intended that sharing experiences would encourage additional thoughts and insights from other group members, and draw out further motivations, opportunities and capabilities, as well as providing groups with the opportunity to make wider connections [32].

### Participants & procedure

We specifically focused on non-profit organisations who were delivering social activities within communities across the UK, including arts, culture or heritage organisations, volunteering organisations, or social or community groups. Thirty-eight representatives of CVSOs took part in six focus groups (group size 3–13) between May and September 2019 lasting 100–120 minutes. A theoretical and purposive approach was taken to sampling [33], to reflect potential differences in barriers and facilitators attributable to region, type of group or activity, or the size, age, or geographical reach of an organisation and groups were held in three different locations around England. Recruitment took place through existing community organisation networks, existing contacts and social media appeals for participants. Travel expenses were reimbursed to enhance ability to attend. No other incentives were offered. The study received ethics approval from the University College London (UCL) ethics committee (14895/002) and all participants provided informed consent. Further detail on methods is available in S1 Material.

A topic guide for conducting the focus groups was developed using the COM-B model as a framework. The guide is presented in S1 Material. Focus groups were recorded and then transcribed in anonymised form. Notes were taken after each focus group, where possible, to capture early ideas about important potential codes, and links between these, and context.

### Data analysis

Reflexive thematic analysis was selected as the analytical approach [34,35]. The process of conducting the thematic approach broadly followed the steps as outlined by Braun and Clarke

[35]: familiarisation with the data, generating initial codes, searching for themes, reviewing themes (verified with a second researcher), defining and naming themes, and producing the report. A combination of inductive and deductive approaches to analysis was implemented: initial coding undertaken in an inductive and open manner, to allow for the codes and themes to be grounded within the data. Coding was undertaken in NVivo qualitative data analysis software; QSR International Pty Ltd. Version 11, 2015. Contradictory data was also included and context around codes was retained. Coding was therefore undertaken at both the semantic and latent level, encompassing not only what has been explicitly expressed by participants, but also interpreted to go beyond description to understanding the barriers and facilitators of the behaviour within the COM-B model.

Codes were then grouped into themes, which represent a "central organising concept" [34]. This stage was undertaken manually. The themes that were generated were discussed with a second researcher during initial development, and at the review stage (as above). These were then mapped to the three domains of the COM-B model: capability, opportunity and motivation. For this project, we interviewed representatives of CVSOs. As such, we analysed our data in relation to the reported psychological and physical capabilities and reflective and automatic motivation of employees working within the CVSOs and the social and physical opportunities open to these CVSOs. Codes, and how these were mapped to themes, along with a theme definition and example data, are presented in the Coding Manual in S1 Material. Following analysis, we applied the Theoretical Domains Framework (TDF), which allows the mapping of specific processes identified by the COM-B model to types of interventions that change behaviour. This allowed us to identify potential interventions that could address the identified barriers and enablers support CVSOs working in this area.

## Findings

### The participants

Participants represented a diverse group of organisations, including grassroots organisations, and national bodies and freelance practitioners, and varied in size and community or cultural focus (see Table 1).

### Themes

Seven primary themes were identified: mental health motivation, community motivation, offering something different, developing skills, understanding boundaries, partnerships, and implementation. The sub-themes, and how these map to the COM-B model are outlined in Fig 1.

**Motivation.** The motivations that were expressed by participants for engaging in this area of work were almost universally positive and enabling. These motivations stemmed from both the personal experience of the participants (having witnessed the benefits of community participation, and often their own experience of mental illness,) along with organisational motivations stemming from improving inclusion in their activities and audiences, and increasing sustainability.

### Mental health motivation

The initial motivation for CVSOs to work with individuals with lived experience of mental illness was often "very much driven by need" *(Participant 3, Focus Group 5)*, prompted by responding to the mental health needs of others, or as a response to the needs of those that come to take part in the service the CVSO offers. Participants were also motivated by wanting

**Table 1. Characteristics of participating organisations.**

| Organisation characteristics | | Number in study (%) |
|---|---|---|
| Type of activity primarily delivered | Arts (incl. choirs, theatre groups and visual arts) | 23 (61) |
| | Community/ Social | 8 (21) |
| | Volunteering | 3 (8 |
| | Heritage | 2 (5) |
| | Libraries | 2 (5) |
| Size of organisation (staff) | 1–9 | 21(55) |
| | 10–50 | 12 (32) |
| | >50 | 5 (13) |
| Region | Scotland | 2 (5) |
| | Wales | 1 (3) |
| | North of England | 2 (5) |
| | East of England | 2 (5) |
| | South West England | 4 (10) |
| | South East England | 9 (24) |
| | London | 9 (24) |
| | UK wide | 9 (24) |
| Strategic importance of working with individuals with mental illness | Primary area of priority | 14 (37) |
| | Secondary (or lesser) area of priority | 24 (63) |

to help others, particularly in the early stages of setting up services or groups, and were often responding to perceived gaps in the already available services. This motivation to help others was very often strongly underpinned by the lived experiences of mental illness of individuals working within CVSOs:

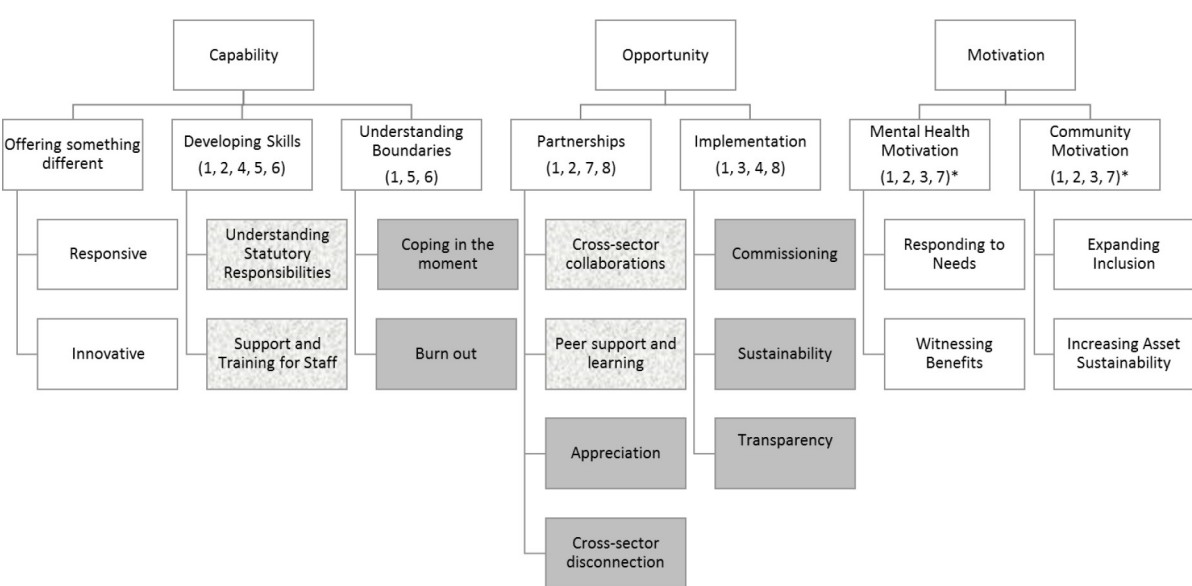

**Fig 1. Themes, sub-themes, and links to COM-B model.** * Whilst these were not barriers to the organisations involved, certain interventions proposed could still help to reduce any motivational barriers amongst other organisations. Grey rectangle, barrier; White rectangle, enabler; Pattered rectangle, both enabler and barrier. Numbers denote links to proposed interventions in Table 2, below.

"I live it all the time, having a foot in both camps actually gives me almost a unique perspective" *(Participant 3, Focus Group 6).*

Observing the benefits of engagement in community activities for their participants was also a strong motivator for participants to continue with their work. Organisations saw that they were making very real changes to their participants' lives, and also to their wider communities:

"they feel held and cared for and seen and acknowledged. . .it gets us up every morning and it gets us moving and changes the world." *(Participant 9, Focus Group 1)*

Some participants expressed that the increase in research evidence for the activities bolstered their motivation to continue: "now it feels like there's a lot of research to show that it is important, it is good, it's helpful for people" *(Participant 2, Focus Group 5).*

### Community motivation

Some participants were motivated to expand their audience diversity, by explicitly trying to appeal to people with lived experience of mental illness who may not wish to engage with mainstream services:

"They were those men who would never have gone to seek traditional support. So I very much wanted to created something that would encourage people like that to seek support" *(Participant 2, Focus Group 6)*

These groups included those who had poor wellbeing as a result of physical illness or children excluded from mainstream schools. Motivation was also drawn from trying to make the arts accessible to those who perhaps could not afford to participate conventionally, and therefore might be excluded from any engagement.

For many of the arts and cultural organisations taking part, the intrinsic value of creativity served as a motivation: expanding access to creative opportunities was fundamental, as was breaking down stigma for their participants, and enabling engagement with their wider communities:

"It was really about integrating people back into community. Addressing stigma, breaking down. . . Giving people a sense of purpose and self-worth" *(Participant 2, Focus Group 2)*

There was a small number of participants who expressed some resistance to the current policy imperatives to work with people with lived experience of mental illness, or at least as a separate activity to their core work. Further, a minority of participants expressed concern about the prescribing of activities changing the way that people value community activities:

"People don't go and do these things usually to cure their mental health and I don't think it should be turned into something that's prescribed. That would take all the joy out of it . . ." *(Participant 3, Focus Group 5)*

The motivation to work with people with lived experience of mental illness originated in part as an organisational strategy to increase sustainability for some organisations. The increased awareness of the extent of mental illness in the population, it was felt, might lead to greater funding and partnership opportunities. This could also mean a more sustainable future, including for organisations that didn't have health as a core focus: "they . . . realised

that, oh soon no-one's really going to care about these dusty old [buildings] that we have. So they've now got this new motto, everyone's in" *(Participant 1, Focus Group 4)*.

For some participants an economic motivation existed alongside the mental health motivations described above. This included accessing funds offered as part of the commissioning process, or grant funding badged for these services. This was acknowledged by one participant to be a motivation which was rarely made explicit:

"We just, well we all deny that we're led by funding which is usually a blatant lie. We're always driven by funding to some extent" *(Participant 6, Focus Group 4)*

The driver of economic motivation not only encouraged organisations to enter this space, but could also shape the services that they might then offer within it. One participant for example discussed their local clinical commissioning group (CCG) indicating that there would be money and a focus on postnatal depression "then yes, we'll do some work on that and maybe get a programme set up for new mums with postnatal depression" *(Participant 1, Focus Group 2)*.

**Capability.** Participants experienced a range of barriers and enablers relevant to their capability. Whilst they discussed the importance of their innovative and responsive services, their ability to deliver these was felt to be restricted by the difficulty of developing the range of needed skills and maintaining boundaries between their activity and therapeutic care, in often small and precariously funded organisations.

## Developing skills

The theme of 'Developing skills' encompassed some barriers, including CVSOs understanding statutory responsibilities and having sufficient support and training for staff, which was particularly a barrier for smaller organisations. They felt less able to recruit and fund teams with diverse skills than their larger counterparts, legislation for safeguarding and General Data Protection Regulation (GDPR) caused concerns around reduced knowledge and confidence and the possible consequences of these (e.g. worries that "we're going to get sued" *(Participant 3, Focus Group 4)*), including acting as a barrier to entering into this type of work with individuals with lived experience of mental illness. For those participants running social enterprises, the demands of running the business, often on a limited budget or staff meant acquiring new skills, and the ability to do this can determine future sustainability of the company, and its ability to provide services.

However, larger organisations found that they were able to fulfil these responsibilities more easily, with larger workforces and more specialist teams, allowing them to work more easily in this area.

"You know, we have a very good HR department which. . .if things do escalate we can refer through" *(Participant 7, Focus Group 3)*.

Concerns around ability to train, and support, staff members, including freelance practitioners, fully and appropriately was a key theme. It was acknowledged that a broad skill set is needed among the wider team, and where this isn't available this can act as a barrier. That the pace of growth of the CVS involvement in working with people with lived experience of mental illness was not matched by the acquisition of the necessary skills was of particular concern:

"It seems that it's picking up steam quite quickly. . .I just think there's a lot in there for people to deal with and it's about finding some sort of proportionate way for different scales of

arts organisations as well to be confident that they can handle what they may have to deal with." (Participant 4, Focus Group 5)

Again, this appeared to be less of a barrier for some of the larger, or more established organisations with separate Human Resources teams or ability to access specialist training and support. However, other organisations shared that they had begun working in this space before thinking about the impact upon their own staff mental health. An ability to provide suitable supervision and training for staff to support their own practice and wellbeing was seen as crucial to building staff skills and ability to provide the service. One participant represented an organisation running courses for musicians working 'in challenging settings' and experiencing 'burn out':

"so you know that's a very specific group and way to kind of support that group. And again it's just really too small, it's just a tiny amount of people per year and it's just not enough" *(Participant 2, Focus Group 5).*

However, it emerged in the groups that some freelance practitioners did not have equitable access to the training and support offered by organisations, and that this impacted on their practice and wellbeing:

"I'm a freelance, employed by them. I'm very solitary. . . I love it but I feel that with this move for social prescribing . . . It's a very unsupported role" *(Participant 4, Focus Group 4).*

## Understanding boundaries

'Understanding boundaries' included further barriers, such as the challenge for organisations coping in the moment delivering projects and staff becoming burnt out. There was a strong resistance among participants to being perceived as a mental health support worker, rather than an artist or community organisation. Being unable to cope in the moment, or maintain these boundaries was perceived as a barrier to the sustainability of working in this area. The sentiment that the work was "therapeutic" but that organisations and their staff are not therapists was expressed by several different participants: "we . . . as an arts organisation, heritage organisation, are not mental health workers. Like that is not the remit of what we're doing." *(Participant 8, Focus Group 3).*

Being able to maintain these boundaries was seen as essential, but very difficult when being asked to cope in the moment with potentially challenging participants and their needs. Along with the possible consequences in the moment for both the practitioner and the attendee, this lack of knowledge and confidence might, in turn, have serious consequences for the success or otherwise of community participation, or social prescribing schemes:

"Because at the end of the day, where does the buck stop? If a GP or a social prescribing facilitator is prescribing, and a patient goes and has a really awful experience then they're not going to even go and try anything else because that will be it." *(Participant 3, Focus Group 2).*

Several participants conveyed a belief that artists and practitioners are particularly vulnerable when asked to draw boundaries. This further links to the predominantly compassionate motivations for entering the sector initially, which are often grounded in lived experience.

Conversely, where the work has arisen from economic motivations, or to expand inclusion, there is deep concern that this has led to an increase in organisations under-appreciating the importance of mental health skills and boundaries between the sectors:

"I get really scared when I hear people say, yes, I'm going to run therapeutic classes. That's very dangerous. You wouldn't say, oh yes, I'll go and help somebody with a broken leg, so why on earth do you feel you can do that with mental health?" *(Participant 2, Focus Group 2).*

Becoming overwhelmed by coping with demands that require expertise in varied areas was a further significant barrier to the continuation of working in this field: "I feel incredibly lonely. At the moment I feel incredibly stressed" *(Participant 2, Focus Group 1).* Restrictions often placed by funders on applying for staff costs contributed to a feeling of being overwhelmed, creating a vicious circle. Further, a lack of wider support for those involved in leading or setting up these organisations contributed significantly to a sense of isolation which in turn led to feelings of being overwhelmed. This was particularly acute for the smaller and newer organisations in the groups. The ability to avoid burn out appeared contingent on having a team of staff to step in, rather than reliance on one or two individuals. The consequences of feeling unsupported and as if the organisation is under threat from lack of funding, are significant personally for those involved, and some participants spoke of the exacerbation of their own mental health problems as a result of this. There was a recognition that this could have further consequences for the effectiveness of the organisation.

## Offering something different

There were clear enablers for CVSOs were around 'offering something different': being able to be responsive to the needs of individuals and offer something innovative as a service.

Participants believed that their organisations' primary strengths were being person-centred and inclusive, rather than a 'one-size fits all' approach to mental health needs, thought to be a significant strength and enabler. The diversity of the sector as a whole was also seen as a particular strength in doing this work: there would be something that could appeal to different people according to their personal tastes, and this would encourage more people to take part: "so I would be less comfortable in a theatre environment but I'm very comfortable in the outdoors and that's fine" *(Participant 2, Focus Group 4).* Several participants could point to examples of how their responsive approach had succeeded where working with people with lived experience, specifically where more conventional approaches had not:

"What runs through it is inclusion and being absolutely person patient, young person centred. That's absolutely at the core of it. And people come to our sessions that don't feel safe to go anywhere else. They come only to us, and that says it, doesn't it?" *(Participant 1, Focus Group 6)*

Staff and volunteers within this field were able to foster connection with potentially vulnerable people, and to demonstrate empathy. They often believed this was not possible to train someone to do and not something that everyone could offer. This ability to empathise was often linked to the lived experience at the heart of many of the organisations, and to further emphasise the distinct nature of this work from that of mental health care. The ability of participants within organisations to spend time with their participants, and to build trust, was also seen as a distinguishing feature:

"What that means is that you just do whatever is best for you. And if we're doing an activity. . . and your baby needs changing . . .It's fine and that there's no judgement around that.". *(Participant 1, Focus Group 2)*

Bringing diverse groups together was seen as a further strength of organisations wishing to work with people with lived experience of mental illness. This was linked to the ability to co-produce services or activities with those that may wish to use or benefit from them. There was a perception that these important, distinctive capabilities, were hard to capture, and therefore hard to measure and demonstrate to funding bodies:

"So you might be planning an activity and delivering something, but there's so much around that to make that happen and to make that a positive experience, which isn't often seen by funders". *(Participant 10, Focus Group 1)*

The distinctive nature of the activity, such as being able to offer creative opportunities, was also perceived to be an enabler and strength of the sector in working with people with lived experience of mental illness, and this was thought to be transformational in its potential. Some of the organisations offered innovative services to address marginalisation and exclusion, such as transport to activities in disconnected areas: "it does mean they turn up. Whereas they might not if they feel a bit alone that day". *(Participant 3, Focus Group 3).*

The distinctive delivery of the activity was also seen as a strength, and this was characterised by the less hierarchical approach offered than that of other sectors. Examples given were of rehearsal techniques that did not place the director at the centre of the work in a theatre group, and enabling the participants to take control of the process. Several referred to working with participants with the aim of increasing their independence and "making them not need me".

There was excitement as well about the potential for organisations such as Community Interest Companies (CICs) (a form of limited company in the UK, where the company aims are for community or society benefit, and profits are donated or reinvested to support that aim) to 'shake things up' with new and different ways of operating outside of the restrictions traditionally placed upon charities.

Participants placed emphasis on increasing their participants' independence, and offering skills and training that would enable them to continue to pursue this activity or others independently in the future. This might specifically be around volunteering and skills, "We try and upskill people. We try and give them training and uniform and experience and a reference and then hopefully send them on their way". *(Participant 4, Focus Group 6)* or the ability of some groups in particular to offer creative opportunities. There was a perception that a benefit of the creative and community sector is that art or creative work produced would be valued separately to the illness or otherwise of the creator and this in turn was felt to have profound consequences:

"When they come back up from the art room with a work of art even if they can't really talk about it or articulate it, they're more human because they made something" *(Participant 1, Focus Group 5).*

**Opportunity.** There were significant barriers in the social and physical environment for CVSOs working with people with lived experience of mental illness. Aspects of the theme 'Partnerships' could be both barriers and enablers depending on whether the partnerships ran well, including cross-sector collaborations and aspects of peer training and learning. But a lack

of appreciation from larger voluntary organisations, charities, or NHS organisations using the services was a barrier to many organisations, as was a feeling of cross-sector (NHS to CVS) disconnection. Within the theme of 'Implementation' concerns around sustainability, and the difficult funding environment were substantial barriers to organisations wishing to work in this area. Developments such as social prescribing were an opportunity but the lack of consistent transparency around how they worked presented an obstacle to further engagement.

## Partnerships

It was clear to participants that for the success of CVSOs working with people with lived experience of mental illness there needed to be collective working with the health sector in particular. In this way this theme represents barriers and enablers. Collaborations would allow the work to flourish, as those with the appropriate skills would bring these to work together: "sometimes you work alongside sort of health specialists but that's quite rare actually for us" *(Participant 4, Focus Group 5).*

But the difficulty of developing these served as a barrier to working in this area. Collaboration was believed to be fundamental to being able to work in this space for nearly all of the participants in the focus group, and in the majority of cases the experience of working in collaboration had been a positive one. Effective partnerships could also provide some support for organisations that otherwise would have experienced some of the isolating effects of working in this space, as outlined above.

It appeared to be vital to a successful collaboration that it was one from which both partners were able to benefit, and learn. There was also a perception amongst some of the grassroots organisations, that there was a lack of support for those working on the frontline, from NHS organisations, or sometimes large charities working in the mental health sector:

"I do struggle a little bit [with] some of the people working in organisations that don't actually work on the front line and don't actually interact". *(Participant 2, Focus Group 6)*

Whilst a collaborative approach and effective collaborative partnerships were seen as essential, these were not always successful, and there was thought to be some skill in knowing when to leave partnerships, as one participant noted: "yes, so not all collaborations are good ones. It's knowing when you go let's walk away from this "*(Participant 1, Focus Group 3).*

Another important opportunity to support this work came through training and learning from peer organisations. Peer group sharing and support took place at the focus groups: many participants took the opportunity to swap contact details, and information about funding and networking events. There was believed to be significant value ("you go away feeling a bit more inspired or a bit more confident" (Participant 2, Focus Group 2)) in the supportive benefits that could be garnered from some peer networks.

Co-location was a key collaborative approach, with many assets being able to use other organisations facilities to host services. This was often fundamental to being able to provide the service at all:

"we do have what we call co-locations with libraries in [location] . . . one or two of my staff are based there to do outreach work where people come in and can have a chat. And we find that to be very useful. The library managers have been very positive." *(Participant 4, Focus Group 1)*

Participants felt, to an extent, exploited by organisations in their own sector who requested services and expertise for free, were keen to use the innovation and risk of smaller or grassroots

groups in order to enter the mental health field themselves. However, they felt equally under-valued by the health sector, and those commissioning services, particularly. This was a further barrier to continuing to work in the field. There was a belief that the time taken to develop the service or activities was unrecognised, and seen to be something that others could just pick up:

> "The other thing is that I get, oh what you did was so amazing, can you write it down so we can do it? If you just tell us your methods, we can go up and do it. It doesn't work like that. They're not recognising the professionalism of our professions." *(Participant 2, Focus Group 2).*

There was a general belief that the health service in particular felt that community provided services that were "free". It was further acknowledged again that perhaps some of the compassionate and benevolent motives that led people to work in the sector, also made them vulnerable to exploitation by organisations with more power wishing to enter the field. One participant noted that she felt "like I'm slightly being exploited in certain relationships. I feel that I'm giving away a lot for free . . . but then the altruistic part of me goes, I want to share and do loads of good. And I think that's really true across the sector." *(Participant 3, Focus Group 2).*

The relationship becomes further complicated, where small organisations are promised that taking part in events, or working for free will bring them benefits in kind. Anecdotes were shared around several instances of these organisations going on to use the ideas and innovations developed by grassroots organisations. In these cases they were often able to then offer the service or activity for free to their clients, or on a larger scale, which was felt to weaken those trying to run as social enterprises. This poses a threat to the sustainability of the smaller grassroots organisations, or freelance practitioners.

A fundamental finding from the focus group data was the feeling of disconnection between the community, cultural and voluntary and health sectors. This barrier was primarily characterised as the two 'speaking a different language', and a 'clash of cultures'. Whilst the sub-theme of appreciation illustrated how the community sector felt like the 'other', this showed that they also regarded the health sector in a similar way, as "speak[ing] a different language" *(Participant 2, Focus Group 2).*

It was acknowledged that both sectors would need to be willing to try to overcome this to allow programmes to work. This area of work was believed to be most successful, and most beneficial to those for whom it was set up, when a range of differing experiences, sectors and types of expertise were able to work together as partners.

## Implementation

Opportunities for commissioning were perceived as being reliant primarily on personal connections. Several examples were offered of commissioning resulting from friendships with commissioners, or other 'ad-hoc' routes. One participant re-counted an incident where a personal connection led to her programme being commissioned: "And I was just thinking, no this is a problem. How do you find arts organisations when you don't have well placed friends?" *(Participant 1, Focus Group 2).*

This was the source of some frustration, and acted as a barrier, for those who did not already have these personal relationships established. There was a further lack of understanding around how to navigate these pathways, even where participants had established contacts, and built relationships.

There was, in addition, an impression that regional differences in willingness to commission the community and voluntary sector were compounding inequalities in access to these: "It's linking up isn't it in networking all those different gaps and some. regions are a lot more open to the arts than others" *(Participant 5, Focus Group 4)*

It was also felt that the lack of shared language outlined above meant that commissioners were unable to understand the processes and benefits involved in taking part in community and cultural activities, and this led to a reluctance to commission these:

Concerns about the sustainability of organisations working in this environment, and the threats the assets faced through lack of funding, or a shortage of the type of funding needed, was a universal theme across all focus groups. Participants often had to address complicated funding criteria, in turn to then only receive small amounts of money:

> "I get fed up of just little bits here and we can run to the end of the year, then we can run for another six months and you know, you've never got that sense of, we've got five years of funding." *(Participant, Focus Group 2)*

It was a source of anxiety for community and voluntary organisations that funding to support staff, either in terms of staff training, or supervision to support their mental health needs was very difficult to obtain. There was further anxiety that having to write funding applications to the funders' requirements resulted in not always being able to fund, and provide, what their participants needed.

A significant concern of the organisations was that they should be able to try and keep the activities free and accessible, to minimise barriers to participation experienced by people with lived experience of mental illness. This was difficult in the light of "need rising and funding falling" as one participant termed the funding environment. Some organisations were able to address this to an extent by using charges or trading to subsidise services for those that wouldn't otherwise be able to afford them.

A further anxiety was expressed around the employment and funding of link workers rather than funding for the activities to which people would be referred. It was perceived that the NHS saw CVS groups as free, without the understanding of the costs to the organisations themselves to provide services, that "this is costing our organisation money" *(Participant 7, Focus Group 3)*. The policy of funding link workers within the NHS compounded this:

> "The challenge is that all that money goes via primary care networks to support link workers. No money to the voluntary sector yet". *(Participant 2, Focus Group 4)*

Participants were therefore aware that the rhetoric around social prescribing was increasing, but felt that this had overtaken the development of the infrastructure for this to work in practice. Many of those who were keen to take part in social prescribing had found that their efforts to do so had been largely unsuccessful; or that the information that they had been given was confusing and contradictory:

> "Someone asked the question we all had which was, I run an arts organisation, how can I get people to be signposted to me by these link workers that you've invested in for your social prescribing plan and no-one [at a conference about social prescribing] could answer". *(Participant 6, Focus Group 4)*

This lack of transparency formed a key barrier to participation in these schemes. Those participants that had been more successful in cultivating links with the health sector or

commissioners in their area often faced barriers from health professionals being reluctant to prescribe or refer to them. Participants voiced fears around how social prescribing would therefore be implemented. The lack of transparency and shared knowledge caused anxiety, and many expressed the worry that there would be no 'quality control' over who was referred to, and that this could create unsafe environments for potentially vulnerable people, or cause people who may otherwise have benefitted to avoid returning to this type of support.

> "Whereas if they'd gone to something else that was a bit more geared up towards supporting that, [people with lived experience of mental illness] they would have had a better experience. So where does the buck stop? Does that then go back to the medical setting stuffed up, or did the choir leader stuff up?" *(Participant 3, Focus Group 2).*

This further linked to doubts around new link workers knowledge of the voluntary sector landscape in their area, and the duplicated effort of "put[ting] a load of money into getting somebody else to come from kind of a health background and then suddenly try and map the cultural sort of assets of another region or locality" *(Participant 1, Focus Group 5)*, repeating work that had already been done by CVS groups.

## Discussion

This study explores barriers and enablers to CVSOs working with individuals with lived experience of mental illness. Notably, the motivations cited by the participants were almost universally positive facilitators of engagement: the participants in the focus groups were highly motivated to take part, with just a small representation of the view that such work might negatively impact on the purpose of CVSOs or the way their work is received. However, the latter view generally reflected that organisations should not be labelling their work, rather than not working with people with lived experience of mental illness. This finding reflects broader evidence of motivation amongst CVSOs to engage with topics of mental health, as evidenced by discussions in recent reports [36,37].

Other clear facilitators for CVSOs included organisational strengths as person-centred, empathetic and safe spaces, offering innovative, creative, and distinctive opportunities, combined with witnessing at first–hand the benefits for their participants, provided ongoing motivation to continue with the work. This echoes findings from qualitative studies that have emphasised the role of these features in enabling mental health benefits for participants [38–40]. Further, strong partnerships, particularly cross-sector links with the health sector, facilitated the capabilities of CVSOs to do this work. For example, successful links supported the organisations with training or supervision in mental health skills, supporting the staff's own wellbeing, and helping them to maintain boundaries.

However, there were several aspects, predominantly in the wider environment ('physical and social opportunities') that were seen as undermining the success of organisations working in this space. The difficult funding environment was a significant barrier. Organisations found funders' demands were often complex, and that grants were short-term and predominantly to fund projects. Restricted public spending in the form of 'austerity' policies has significantly impacted the charity and voluntary sector [41], reducing the funding available to support training, supervision, and the long-term delivery of services [42,43]. This in turn has made it harder for CVSOs to maintain or navigate important boundaries, leading to participants feeling overwhelmed. Indeed, burn-out has been much discussed in the delivery of health services [44] and can impact upon skills in care-giving and staff turnover (two other barriers identified through this study). Further, the distinctive and innovative offer of some of the organisations,

recognised as a strength in their work with people with lived experience of mental illness, sometimes conversely meant that their success was hard to capture in the measurements and language that would appeal to fund holders and commissioners.

Additionally, this study identified that commissioning pathways can lack transparency, and there was a sense that commissioning opportunities often emerged as a result of friendships and personal connections. Accountability and transparency in the commissioning process have become increasingly fragmented with recent NHS policy developments [45–47]. The lack of commissioning pathways, and difficulties in obtaining funding for aspects of the work that organisations actually need, such as staff, and staff development, were identified as significant barriers to continuing to work in this field. Altruistic motivations such as wanting to help, often rooted in own lived experience, meant organisations felt more vulnerable to the exploitation of their skills and creativity, as they in turn wished to help others. Problems with larger organisations taking advantage of this (reflected more broadly by relationships with commissioners [48]) in turn threatened sustainability. Feeling under-valued as a sector had recently been compounded by social prescribing policy not including additional funding for the community organisations that would actually be providing the services. There were additional concerns around social prescribing implementation, such as quality control. A recent report on quality assurance in social prescribing schemes also highlights the needs for awareness and skills for providers around understanding and enacting their statutory responsibilities [49]. These concerns linked to organisations entering this space in order to increase their own sustainability, which raised unease that providing a safe and effective environment would not be prioritised. It was felt that this may then damage the process and community participation in general.

It is clear, therefore, that interventions are needed to address these barriers in order that high quality community programmes can be delivered to those who could benefit. It is recommended that interventions that seek to address behaviour change at an individual or organisational level are designed by using a rational system that covers all possible intervention types, matching appropriate interventions to features of the target population, the context, and the specific barriers identified[30]. The behaviour change wheel (BCW) is one such framework, which draws links between the COM-B framework and different intervention functions[30]. Mapping the COM-B barriers identified in this study onto the BCW highlights several types of interventions that could support CVSOs to engage with individuals with mental illness moving forwards. For example, behaviour change techniques such as incentivisation and restructuring the physical environment [30,50] would allow the establishment of long term funding that could support on-going work, as well as new projects, and to include core staff costs. This would enable CVSOs to sustainably fund staff training and supervision to support staff wellbeing. The proposed interventions derived from the data are detailed in Table 2, below:

This study had a number of strengths. There was strong representation from a variety of organisations from across the sector, ranging from freelancers to large organisations, based in urban and rural settings of differing levels of deprivation in different parts of the country, covering a range of community activities including arts, heritage sites, social clubs and outdoors activities. The research was guided by an established theoretical framework and our use of multiple focus groups enabled us to confirm and explore themes in depth. Further, these focus groups provided benefits besides the purpose of data collection: participants had an opportunity to network, share practice and concerns and funding information, and positive feedback was received from participants about the process. However, there were several limitations. Some smaller organisations were unable to participate due to the pressure of work so the barriers faced by these groups with smaller resources may have been less represented. Further, the self-selection of participants into the research meant that those CVSOs who participated were

**Table 2. Proposed interventions to promote increased engagement with people with lived experience of mental illness of CVSOs, linked to behaviour change techniques.**

| Intervention Number | Number of barriers that could be addressed[a] | Intervention type | Behaviour change techniques | Outline of strategy |
|---|---|---|---|---|
| 1 | 5 | Enablement | Social support | Buddy/ mentoring schemes for local organisations to support new organisations in working with health organisations/commissioners/link workers |
| 2 | 4 | Enablement / environmental restructuring | Social support / restructuring the social environment | Establishment of local cooperatives of CVSOs who can work to develop partnerships with the health sector as a collective, sharing tasks such as developing contracts and business plans and organising/paying collectively for training, and provide informal support |
| 3 | 3 | Environmental restructuring / incentivisation | Restructuring the physical environment / material reward | Provision of long-term funding from funders to support both new and ongoing work, and to include core costs including the support of staff training wellbeing |
| 4 | 2 | Training | Instruction on how to perform the behaviour; demonstration of the behaviour; Behavioural practice or rehearsal; self-monitoring of the behaviour | Training guides or events to be available to CVSOs on organisational issues such as how to engage with commissioning process and how to build a business case around working with health, with follow-up sessions to monitor implementation of learning |
| 5 | 2 | Training | Instruction on how to perform the behaviour; demonstration of the behaviour; Behavioural practice or rehearsal; self-monitoring of the behaviour | Training guides or events to be available to CVSOs on staff skills such as how to work with individuals with specific mental health needs and how to safeguard individual wellbeing, with follow-up sessions to monitor implementation of learning |
| 6 | 2 | Enablement | Goal setting; Review behaviour/ outcome goals; verbal persuasion about capability | Provision of template partnership guidelines for CVSOs and health organisations (i) providing templates for initial meetings between partners that support sharing of organisational mission statemenets and capabilities, clarify language, and set boundaries of expertise; (ii) providing template evaluations for organisations to set their own goals on all aspects of projects (including partnership working, staff coping, organisational capabilities) and review progress at specific milestones |
| 7 | 2 | Enablement / modelling | Information about others' approval / demonstration of the behaviour | Sharing of case studies of good practice in partnership working and evaluations and positive feedback showing the mutual appreciation of sectors in successful partnerships |
| 8 | 1 | Environmental restructuring | Restructuring the physical environment | Development of simplified, clearer and more standardised processes within schemes such as social prescribing for link workers to engage with community organisations |

[a] The specific barriers that can be addressed by each intervention in the table are shown in Fig 1

probably already engaged both in the work, and this is possibly reflected in the motivations for entering the field being almost universally positive.

## Conclusion

This study is the first to explore the barriers and facilitators to working with people with lived experience of mental illness for CVSOs. It highlights that there are strong motivations for CVSOs to work with individuals with lived experience of mental illness, as well as other facilitators including certain organisational strengths, good partnership working, and experience of

the benefits to participants. However, there are also a number of barriers, particularly relating to the wider environment including issues relating to funding, training, and sustainability. There are a range of interventions that could to address these barriers. Some of these could be simple to establish and simultaneously address multiple barriers, such as networks and cooperatives to support aspects of organisational development, partnership building, and business-related tasks. However, other interventions require higher-level buy-in and more complex changes, such as modifications to the types of funding available for CVSOs, clarification of processes within schemes such as social prescribing, and provision of training on working in mental health. Future studies are encouraged that could develop the potential interventions identified here further, including designing, delivering and testing the effectiveness of specific interventions. This work is crucial if we are to ensure that individuals with lived experience of mental illness have equal opportunities to engage in community activities.

## Supporting information

**S1 Material.**
(DOCX)

## Acknowledgments

The authors wish to thank the participating organisations. They would also like to thank Ms Vas James, Ms Hei Wan Mak, and Ms Sophie Large for valuable support with moderating the focus groups.

## Author Contributions

**Conceptualization:** Louise Baxter, Daisy Fancourt.

**Data curation:** Louise Baxter.

**Formal analysis:** Louise Baxter, Daisy Fancourt.

**Investigation:** Louise Baxter.

**Methodology:** Daisy Fancourt.

**Writing – original draft:** Louise Baxter, Daisy Fancourt.

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
