## [Decision Letter · Decision Letter 0]

26 Mar 2020

PONE-D-19-34366

What are the barriers to, and enablers of, working with people with lived experience of mental illness amongst community and voluntary sector organisations? A qualitative study

PLOS ONE

Dear Dr Fancourt,

Thank you for submitting your manuscript to PLOS ONE. After careful consideration, we feel that it has merit but does not fully meet PLOS ONE’s publication criteria as it currently stands. Therefore, we invite you to submit a revised version of the manuscript that addresses the points raised during the review process.

Thank you for submitting your manuscript to PLOS ONE and my apologies for the length of time it took to secure peer reviews. As you will note, however, once we found the reviewers, they were highly positive about your manuscript and believe it will have an important impact in the field. The qualitative methods were well reported and reviewers request only a few additional details. 

The major comment was that this manuscript could be shortened by tightening up the presentation of the results. Figure 1 does an excellent job of communicating the main themes in the context of the theoretical framework. I might suggest editing the Results section so that findings are presented by the 7 main themes only and integrating discussion of the sub-themes within these sections. Your interpretations are well-evidenced, however, reviewers suggested they may have even greater impact by carefully selecting illustrative quotations and editing these to ensure they are concise and clear.

I look forward to receiving a revised version of the manuscript. Be well in these chaotic times.

We would appreciate receiving your revised manuscript by May 10 2020 11:59PM. To enhance the reproducibility of your results, we recommend that if applicable you deposit your laboratory protocols in protocols.io, where a protocol can be assigned its own identifier (DOI) such that it can be cited independently in the future. For instructions see: http://journals.plos.org/plosone/s/submission-guidelines#loc-laboratory-protocols

We look forward to receiving your revised manuscript.

Kind regards,

Quinn Grundy, PhD, RN

Academic Editor

PLOS ONE

Journal Requirements:

Reviewers' comments:

Reviewer's Responses to Questions

**Comments to the Author**

1. Is the manuscript technically sound, and do the data support the conclusions?

Reviewer #1: Yes

Reviewer #2: Yes

Reviewer #3: Yes

2. Has the statistical analysis been performed appropriately and rigorously? 

Reviewer #1: Yes

Reviewer #2: N/A

Reviewer #3: N/A

3. Have the authors made all data underlying the findings in their manuscript fully available?

Reviewer #1: Yes

Reviewer #2: Yes

Reviewer #3: Yes

4. Is the manuscript presented in an intelligible fashion and written in standard English?

Reviewer #1: Yes

Reviewer #2: Yes

Reviewer #3: Yes

5. Review Comments to the Author

Reviewer #1: This is an interesting paper, which describes a qualitative study to understand the motivations and barriers to working in community and voluntary sector in supporting participants with mental illness.

Overall the narrative in clear, logical and well presented, although the results section is very lengthy and would benefit from being made more concise, to improve readability. For example, many of the quotes are paragraphs long and could be cut down to retain the key points. Some of the quotes would also benefit from being edited for clarity, possibly a mismatch between speech and typed communication? (suggestions outlined below).

I have a few minor recommendations, mainly small points which would be clearer/would benefit from more explanation:

Line 55. What are the positive psychological factors, can you discuss briefly?

Line 62. ‘for older people’- does this apply to all three types of community projects, or just community theatre?

Line 72. Could you elaborate on the evidence and reasons for the increasing demand? Better diagnosis, more interventions, higher levels of mental illness, etc?

Line 96. Is the ‘B’ in COM-B short for ‘Behaviour’? Please specify.

Line 147. How was the grouping done, manually or through a code? A little more explanation would be helpful.

Table 1. ‘Type of activity primarily deliver’- typo?

Table 1. Please ensure all percentages add up to 100.

Line 171. Where is this asterix referring to?

Line 214. “... research to back that up” is unclear, could you add some context in [brackets]?

Line 306. “Larger organisations found they were able to fulfil these responsibilities”- evidence needed.

Line 354. “Yes we as, well if we start the project we as an arts organisation...”. This line doesn’t make sense.

Line 426. “responsive approach had succeeded where in working with people”- typo?

Lines 436-439. Assume SP is a person, but no explanation for acronym or numbering given, so this is confusing to the reader. (Also Line 632, 674 etc)

Line 460. “one participant did raise some ways”- please explain.

Line 478. I think there’s a double negative here, should this be,”does”?

Line 510. Appreciation by who?

Line 510. Please explain “cross-sector disconnection”.

Line 561. Quotation within quotation needed.

Line 570. Provide training to who?

Line 730. “I think we’ve got this green, you know and a very unused to thinking outside the box and hoping that might change”, isn’t clear at all what they are trying to say.

Line 737. First sentence of the quote- supporting what? More context, please.

Line 798. More explanation needed for the ‘COM-B Wheel’. What is this and how has it helped you derive in interventions/strategies?

Table 2. Please specify the barriers.

Intervention 6. “between partners hat”- typo?

Reviewer #2: Dear authors,

Thank you for your manuscript. This was a very interesting piece of research highlighting some key issues facing the voluntary sector at a pivotal time (social prescribing).

Generally, I felt the manuscript was too long. I appreciate the journal does not specify a word count but, to support readability, greater conciseness is needed in the presentation of findings. I have made specific comments below for the authors to consider.

Abstract:

No mention of Theoretical Domains Framework in abstract. Check for typos and spacing.

Background:

Missing literature on voluntary sector services for people with learning disabilities (which could be informative here). Also missing literature on such interventions as care farms for people with mental illness (which could also be information here).

Methods:

Ln 110: “Focus groups were chosen as the data collection tool.” Suggest you remove the term ‘data collection tool’ and integrate this with a statement justifying selection of this method e.g. Focus group were selected in order to….

Ln 111: “This allowed the inclusion of a larger number of organisations than individual interviews”. I do not consider this to be a valid reason. One could interview any number of representatives from any number of organisations. The authors discussion of the type and value of data is more appropriate (e.g. interactions, shared understandings between participants from different organisations).

Ln 121: Typo ‘,.’

More clarity is needed on who was responsible for data collection and transcription (who did what?). Notes were taken to capture early ideas about what? Could the authors be more specific?

I am a little confused about the approach to analysis. On Ln 131 a framework is described. But on Ln 139 the authors describe beginning with an inductive approach. They have stated inductive and deductive approaches were used (which I am pleased to see), but more clarity is needed on which came first. At what point did COM-B inform the analysis?

No information is given on the size of focus groups. What was the average size and range?

Findings:

Table 1: Typo ‘Type of activity primarily deliver’ (delivered). Organisations based in Scotland are described in the table but not in the ‘Participants and procedures’. Can you assure us that you had ethics approval to recruit from Scotland?

Focus group extracts are not labelled with an identifier. This information is important to enable readers to assess whether data from a range of participants is represented in analysis.

Figure 1 is great. Really clear and helpful presentation.

The extract on Ln 250 strikes a really important tone.

Really interesting to note that some organisations were motivated to work with people with mental illness for strategic purposes and to support their own sustainability. Very honest and pragmatic. To what extent is this square pegs in round holes? How far were these organisations deviating from their founding purposes?

Ln 265: Can one be altruistic and economically motivated? Regardless, I am a bit of a cynic when it comes to altruism. If someone is passionate about helping others and they derive a sense of self-worth from this, can their actions be considered truly altruistic i.e. self-less? Also, people operating in the voluntary sector are not not necessarily volunteers, many take a wage (altruistic?). Suggest reframing/rephrasing as: benevolent or compassionate (although I accept a participant uses the term ‘altruism’ on Ln 599).

Ln 290: Typo ‘for’ (For)

Ln 303: Can the authors provide clarity on what CIC stands for?

Ln 388: Check indentation of quote

Ln 404: Check indentation of quote

Ln 414: “organisations primary strengths” missing an apostrophe

Overall, the findings section is quite long. There are a lot of themes and sub-themes to present. I suggest the authors review the level of interpretation and description between data extracts. Whilst this is helpful, perhaps for the sake of brevity this could be reduced.

Discussion:

Ln 813: Weaknesses should be limitations.

Conclusion:

Could be strengthened. What recommendations can be formed from the results? Can you be more specific about the interventions included in Table 2? Some are (arguably) more actionable than others. Did you present these interventions back to the participants? If so, what were their thoughts?

Reviewer #3: This is an interesting and important qualitative study, carried out according to a clear theoretical framework and aims and generating ideas for interventions following appropriate involvement of stakeholders. It is very well written and easy to read. Conclusions are clearly and appropriately drawn whilst outlining the complexity of such an exercise involving heterogeneous stakeholders.

6. PLOS authors have the option to publish the peer review history of their article (what does this mean?). If published, this will include your full peer review and any attached files.

Reviewer #1: No

Reviewer #2: Yes: Dr Tom Kingstone

Reviewer #3: No

---

## [Author Response · Author response to Decision Letter 0]

27 Apr 2020

[Included in the 'Response to Reviewers'

Thank you very much for your comments, and for those from the reviewers, regarding our paper ‘What are the barriers to, and enablers of, working with people with lived experience of mental illness amongst community and voluntary sector organisations? A qualitative study’. We have outlined how we have responded to these below. 

The major comment was that this manuscript could be shortened by tightening up the presentation of the results. Figure 1 does an excellent job of communicating the main themes in the context of the theoretical framework. I might suggest editing the Results section so that findings are presented by the 7 main themes only and integrating discussion of the sub-themes within these sections. Your interpretations are well-evidenced, however, reviewers suggested they may have even greater impact by carefully selecting illustrative quotations and editing these to ensure they are concise and clear.

We are delighted that the reviewers are so positive about the manuscript. We really appreciate the comments we have received and have addressed all of them. In terms of the Results section, this has been edited to reflect the comments above. The sub-theme explanations have been integrated into the main themes, and the illustrative quotes have been reduced and edited. 

Reviewer 1

Overall the narrative in clear, logical and well presented, although the results section is very lengthy and would benefit from being made more concise, to improve readability. For example, many of the quotes are paragraphs long and could be cut down to retain the key points. Some of the quotes would also benefit from being edited for clarity, possibly a mismatch between speech and typed communication? (suggestions outlined below).

We are pleased that Reviewer 1 found the paper clear and well presented, although we appreciate the point that it was previously long. As a result, and in line with the editor’s comments, the findings section has been edited, and the quotes edited for clarity, so it is now substantially shorter. 

Line 55. What are the positive psychological factors, can you discuss briefly?

Added ‘such as life satisfaction and mental wellbeing’

Line 62. ‘for older people’- does this apply to all three types of community projects, or just community theatre?

The community theatre project was for older participants – this has now been clarified in the text.

Line 72. Could you elaborate on the evidence and reasons for the increasing demand? Better diagnosis, more interventions, higher levels of mental illness, etc? 

Clarification added: ‘which has been precipitated by a variety of influences, including workforce shortages, social factors such as unemployment, and welfare restructuring under austerity policies.’

Line 96. Is the ‘B’ in COM-B short for ‘Behaviour’? Please specify. 

The (B) has been added here for clarity.

Line 147. How was the grouping done, manually or through a code? A little more explanation would be helpful.

We have clarified that codes that were similar to one another were then grouped into themes.

Table 1. ‘Type of activity primarily deliver’- typo?

Corrected.

Table 1. Please ensure all percentages add up to 100.

Corrected.

Line 171. Where is this asterix referring to?

The asterix refers to figure 1, which is in a separate file. 

Line 214. “... research to back that up” is unclear, could you add some context in [brackets]?

Added [that their activities were effective]

Line 306. “Larger organisations found they were able to fulfil these responsibilities”- evidence needed.

During the editing process, this section has now been removed.

Line 354. “Yes we as, well if we start the project we as an arts organisation...”. This line doesn’t make sense.

Edited to make sense.

Line 426. “responsive approach had succeeded where in working with people”- typo?

Corrected

Lines 436-439. Assume SP is a person, but no explanation for acronym or numbering given, so this is confusing to the reader. (Also Line 632, 674 etc)

This stands for ‘Speaker’ where more than one speaker’s lines are included in the quote. Quotes have now been edited, so this has been removed.

Line 460. “one participant did raise some ways”- please explain.

During the editing process, this section has now been removed.

Line 478. I think there’s a double negative here, should this be, ”does”?

Corrected.

Line 510. Appreciation by who?

Added: ‘from larger voluntary organisations, charities, or NHS organisations using the services’ 

Line 510. Please explain “cross-sector disconnection”.

Added: ‘NHS to CVS’ to clarify.

Line 561. Quotation within quotation needed.

Quotation marks added

Line 570. Provide training to who?

Added: ‘either to charities, or NHS organisations’

Line 730. “I think we’ve got this green, you know and a very unused to thinking outside the box and hoping that might change”, isn’t clear at all what they are trying to say.

Edited for clarity.

Line 737. First sentence of the quote- supporting what? More context, please.

Added ‘[people with lived experience of mental illness]’ for more context.

Line 798. More explanation needed for the ‘COM-B Wheel’. What is this and how has it helped you derive in interventions/strategies?

We have now clarified that it is recommended that interventions that seek to address behaviour change at an individual or organisational level are designed by using a rational system that covers all possible intervention types, matching appropriate interventions to features of the target population, the context, and the specific barriers identified. The behaviour change wheel (BCW) is one such framework, which draws links between the COM-B framework and different intervention functions. 

Table 2. Please specify the barriers.

The specific barriers are shown in Figure 1, which also shows numbers beside each barrier corresponding to the numbers in Table 2. We have now clarified this in the notes to Table 2.

Intervention 6. “between partners hat”- typo?

Corrected

Reviewer 2

Generally, I felt the manuscript was too long. I appreciate the journal does not specify a word count but, to support readability, greater conciseness is needed in the presentation of findings. I have made specific comments below for the authors to consider.

We are grateful for Reviewer 2’s comments below. We agree that the paper was long so the findings section has now been edited and the manuscript is now substantially shorter.

Abstract:

No mention of Theoretical Domains Framework in abstract. Check for typos and spacing.

TDF added, and spacing corrected

Background:

Missing literature on voluntary sector services for people with learning disabilities (which could be informative here). Also missing literature on such interventions as care farms for people with mental illness (which could also be information here).

We have added literature on care farms for people with mental illness and learning disabilities to the Background section.

Methods:

Ln 110: “Focus groups were chosen as the data collection tool.” Suggest you remove the term ‘data collection tool’ and integrate this with a statement justifying selection of this method e.g. Focus group were selected in order to….

This has been changed, and integrated with the reasons for focus groups, as below.

Ln 111: “This allowed the inclusion of a larger number of organisations than individual interviews”. I do not consider this to be a valid reason. One could interview any number of representatives from any number of organisations. The authors discussion of the type and value of data is more appropriate (e.g. interactions, shared understandings between participants from different organisations).

The sentence referring to larger numbers has been removed. 

Ln 121: Typo ‘,.’

Corrected.

More clarity is needed on who was responsible for data collection and transcription (who did what?). 

In the interest of shortening the manuscript, this detail is provided in the supplementary material. 

Notes were taken to capture early ideas about what? Could the authors be more specific?

Added: ‘about important potential codes, and links between these’

I am a little confused about the approach to analysis. On Ln 131 a framework is described. But on Ln 139 the authors describe beginning with an inductive approach. They have stated inductive and deductive approaches were used (which I am pleased to see), but more clarity is needed on which came first. At what point did COM-B inform the analysis?

The COM-B framework was used as a basis for development of the topic guides, as in line 131. The initial coding was inductive, and then ‘The codes were then grouped into themes, which represent a “central organising concept” (34), and mapped to the three domains of the COM-B model: capability, opportunity and motivation.’ (lines 151-152).

No information is given on the size of focus groups. What was the average size and range

This is now provided in the methods section

Findings:

Table 1: Typo ‘Type of activity primarily deliver’ (delivered). Organisations based in Scotland are described in the table but not in the ‘Participants and procedures’. Can you assure us that you had ethics approval to recruit from Scotland?

The text has been amended to reflect the participation from Scotland. The ethics approval did not contain restrictions to recruitment limiting to England and Wales, and the approval included approaching organisational members of the MARCH network to take part. MARCH covers organisations across the UK, including Scotland. 

Focus group extracts are not labelled with an identifier. This information is important to enable readers to assess whether data from a range of participants is represented in analysis.

Identifiers have been added to the quotes (Participant number, focus group number)

Figure 1 is great. Really clear and helpful presentation. The extract on Ln 250 strikes a really important tone. Really interesting to note that some organisations were motivated to work with people with mental illness for strategic purposes and to support their own sustainability. Very honest and pragmatic. To what extent is this square pegs in round holes? How far were these organisations deviating from their founding purposes?

Ln 265: Can one be altruistic and economically motivated? Regardless, I am a bit of a cynic when it comes to altruism. If someone is passionate about helping others and they derive a sense of self-worth from this, can their actions be considered truly altruistic i.e. self-less? Also, people operating in the voluntary sector are not not necessarily volunteers, many take a wage (altruistic?). Suggest reframing/rephrasing as: benevolent or compassionate (although I accept a participant uses the term ‘altruism’ on Ln 599).

Corrected to ‘mental health motivation’, to reflect the title of the theme this is referring to. Changed from altruistic later in the text. 

Ln 290: Typo ‘for’ (For)

Corrected 

Ln 303: Can the authors provide clarity on what CIC stands for?

The following text has been added for clarity: ‘[Community Interest Company, a form of limited company in the UK, where the company aims are for community or society benefit, and profits are donated or reinvested to support that aim]’

Ln 388: Check indentation of quote

Corrected.

Ln 404: Check indentation of quote

Corrected.

Ln 414: “organisations primary strengths” missing an apostrophe

Corrected

Overall, the findings section is quite long. There are a lot of themes and sub-themes to present. I suggest the authors review the level of interpretation and description between data extracts. Whilst this is helpful, perhaps for the sake of brevity this could be reduced.

Discussion:

Ln 813: Weaknesses should be limitations.

Corrected

Conclusion:

Could be strengthened. What recommendations can be formed from the results? Can you be more specific about the interventions included in Table 2? Some are (arguably) more actionable than others. Did you present these interventions back to the participants? If so, what were their thoughts?

We have now built substantially on this concluding paragraph, providing a summary of findings, highlighting which interventions are relatively simple to implement and which require more work, and making recommendations for how this project could be taken forward. We have not presented the interventions back to participants yet: this paper is intended to move such discussion forwards.

Reviewer #3: This is an interesting and important qualitative study, carried out according to a clear theoretical framework and aims and generating ideas for interventions following appropriate involvement of stakeholders. It is very well written and easy to read. Conclusions are clearly and appropriately drawn whilst outlining the complexity of such an exercise involving heterogeneous stakeholders.

We are pleased that Reviewer 3 thinks this is an important study.

---

## [Decision Letter · Decision Letter 1]

27 May 2020

PONE-D-19-34366R1

What are the barriers to, and enablers of, working with people with lived experience of mental illness amongst community and voluntary sector organisations? A qualitative study

PLOS ONE

Dear Dr. Fancourt,

Thank you for submitting your manuscript to PLOS ONE. After careful consideration, we feel that it has merit but does not fully meet PLOS ONE’s publication criteria as it currently stands. Therefore, we invite you to submit a revised version of the manuscript that addresses the points raised during the review process.

Thank you for your thoughtful and comprehensive revision. The reviewers felt that you well-addressed the bulk of their comments, but had just a few outstanding suggestions. Please address Reviewer 1's question regarding how themes were grouped and consolidated. Please also give the paper a thorough proof read, with attention to punctuation and spacing issues in the Results. Note that block quotations should be used for quotations >40 words.

Finally, although you have reduced the manuscript by over 2000 words, I would suggest further tightening up the results section. You might consider integrating parts of quotations into sentences rather than providing a block quotation for every insight. Or, in introducing sections, rather than listing the themes, provide a sentence that imparts the overall 'take-home' interpretation for that section so it reads as a narrative rather than simply a roadmap. I feel that further consolidating these results will clearly distill the important findings rather than cutting or losing any findings. I would aim to cut another 750-1000 words.

I look forward to receiving a revised manuscript.

We look forward to receiving your revised manuscript.

Kind regards,

Quinn Grundy, PhD, RN

Academic Editor

PLOS ONE

Reviewers' comments:

Reviewer's Responses to Questions

**Comments to the Author**

1. If the authors have adequately addressed your comments raised in a previous round of review and you feel that this manuscript is now acceptable for publication, you may indicate that here to bypass the “Comments to the Author” section, enter your conflict of interest statement in the “Confidential to Editor” section, and submit your "Accept" recommendation.

Reviewer #1: (No Response)

Reviewer #2: All comments have been addressed

2. Is the manuscript technically sound, and do the data support the conclusions?

Reviewer #1: Yes

Reviewer #2: Yes

3. Has the statistical analysis been performed appropriately and rigorously? 

Reviewer #1: Yes

Reviewer #2: N/A

4. Have the authors made all data underlying the findings in their manuscript fully available?

Reviewer #1: Yes

Reviewer #2: Yes

5. Is the manuscript presented in an intelligible fashion and written in standard English?

Reviewer #1: Yes

Reviewer #2: Yes

6. Review Comments to the Author

Reviewer #1: I was pleased to see that most of the Reviewers’ comments have been adequately addressed, although there are two key areas which I feel are still outstanding.

1) Line 151: My query around the grouping of codes into themes has not been addressed. What does ‘similar to one another’ mean? How was this validated? Also, please specify and describe if this was manual or computerised.

2) Quotes from participants in particular are now much clearer, but the results section is still incredibly long, as the quotes are very lengthy and make it difficult to read. It appears that despite the Editors and Reviewers’ comments in the previous round, the length doesn’t appear to have changed. I would suggest selecting the most relevant quotes and editing the section to make it a more manageable length.

Also, a very minor issue, several times the quotes have been removed (lines 447 and 567, in the Tracked document for example), but the colons are still there. Please give the document a good proofread, as there have been several changes, and in particular double check for these discrepancies.

Reviewer #2: Dear authors

Thank you for submitting your manuscript with changes. I have now read through your manuscript and am satisfied that you have responded to comments and suggestions.

This manuscript is much improved and I am recommending that it be accepted for publication, subject to thorough checking of typos.

Kind regards

7. PLOS authors have the option to publish the peer review history of their article (what does this mean?). If published, this will include your full peer review and any attached files.

Reviewer #1: No

Reviewer #2: No

---

## [Author Response · Author response to Decision Letter 1]

12 Jun 2020

Dear Editor,

Thank you very much for your comments, and for those from the reviewers, regarding our paper ‘What are the barriers to, and enablers of, working with people with lived experience of mental illness amongst community and voluntary sector organisations? A qualitative study’. We are pleased that the reviewers felt the comments had been well addressed. We have now responded to the final comments:

1) Line 151: My query around the grouping of codes into themes has not been addressed. What does ‘similar to one another’ mean? How was this validated? Also, please specify and describe if this was manual or computerised.

We have now clarified that we grouped codes into themes and validated this with a second researcher as per our method for identifying codes. We have also clarified that this was manual.

2) Quotes from participants in particular are now much clearer, but the results section is still incredibly long, as the quotes are very lengthy and make it difficult to read. It appears that despite the Editors and Reviewers’ comments in the previous round, the length doesn’t appear to have changed. I would suggest selecting the most relevant quotes and editing the section to make it a more manageable length.

We have further reduced the length of the manuscript, cutting several whole quotes, cutting down other quotes, and integrating short quotes into sentences. We also took the advice of the editor to change the text at the start of each section of the results so that we summarise the findings in a ‘take-home’ message. We do not feel we can make any further cuts without leaving points unsubstantiated, but as the manuscript is now significantly shorter than it was at submission, we hope it is now acceptable.

Also, a very minor issue, several times the quotes have been removed (lines 447 and 567, in the Tracked document for example), but the colons are still there. Please give the document a good proofread, as there have been several changes, and in particular double check for these discrepancies.

We have now carefully read the manuscript again and corrected any errors we spotted. We will also ensure that we do another thorough proof-read at the proof stage.

With all of these changes made, we now hope the manuscript is suitable for acceptance.

Yours Sincerely,

Louise Baxter

Research Fellow in Mental Health

Research Department of Behavioural Science and Health

Institute of Epidemiology & Health Care

University College London

---

## [Editor Report · Decision Letter 2]

15 Jun 2020

What are the barriers to, and enablers of, working with people with lived experience of mental illness amongst community and voluntary sector organisations? A qualitative study

PONE-D-19-34366R2

Dear Dr. Fancourt,

We’re pleased to inform you that your manuscript has been judged scientifically suitable for publication and will be formally accepted for publication once it meets all outstanding technical requirements.

Kind regards,

Quinn Grundy, PhD, RN

Academic Editor

PLOS ONE
---

## [Editor Report · Acceptance letter]

23 Jun 2020

PONE-D-19-34366R2 

What are the barriers to, and enablers of, working with people with lived experience of mental illness amongst community and voluntary sector organisations? A qualitative study 

Dear Dr. Fancourt:

I'm pleased to inform you that your manuscript has been deemed suitable for publication in PLOS ONE. Congratulations! Your manuscript is now with our production department. 

Kind regards, 

on behalf of

Dr. Quinn Grundy 

Academic Editor

PLOS ONE